# Measuring Importance of Physical Self-Care Behavior in Patients with Heart Failure: Validation and Reliability Analysis of 14-item IPSC Scale

Athanasia Tsami *, Ioannis Koutelekos, Georgia Gerogianni, Niki Pavlatou, Antonia Kalogianni, Theodore Kapadohos , Georgia Toulia and Maria Polikandrioti

Department of Nursing, University of West Attica, 12243 Athens, Greece
* Correspondence: athtsami@uniwa.gr

**Abstract:** Introduction: Heart failure (HF) is a complex clinical syndrome associated with increased disability, morbidity and mortality globally. HF is characterized by recurrent exacerbations and a high rate of hospital readmissions. Self-care is a crucial component of treatment. The way patients assess the importance of self-care may shed light on planning effective individualized interventions. The aim of this study was to conduct a validity and reliability analysis of the new 14-item IPSC scale, which measures how important HF patients consider their physical self-care behavior (IPSC, Importance of Physical Self-Care). Material and Methods: The 14-item IPSC scale was created by the researchers to explore how important HF patients consider their physical self-care behavior. The validation of the IPSC scale included face and content validity, construct validity, internal consistency, repeatability and discriminant validity. Patients' characteristics were also included. Results: In the present study, 52 hospitalized HF patients were enrolled, of whom 55.8% were female, 34.6% above 70 years old, 48.1% of NYHA class III and 32.7% suffered this illness from 6 to 10 years. The total IPSC score had a possible range of 14–56, with higher scores indicating a low importance of self-care. The descriptive statistics in the IPSC scale showed a mean score of $27.9 \pm 4.9$ and a median score of 29, indicating that HF patients evaluated self-care behavior as moderately important. All items were found to be statistically significantly correlated with total scale scores ($p \leq 0.05$), with correlation coefficients rho > 0.250, indicating moderate to strong correlations and meaning that all items are important in the calculation of the final score (construct validity). The internal consistency of the items that constituted the total score was found to be high (Cronbach's a > 0.7). Furthermore, it was found that scores had high repeatability ($p \leq 0.001$ and ICCs > 0.7). Regarding discriminant validity, a statistically significant association was observed between the importance of physical self-care behavior and both years suffering the illness ($p = 0.007$) and the NYHA class ($p = 0.030$). Conclusion: The 14-item IPSC scale is a reliable instrument that help nurses in clinical settings to gain a better and prompt understanding of the importance which patients invest in their physical self-care behavior.

**Keywords:** heart failure; self-care; validation; reliability

## 1. Introduction

The clinical syndrome of heart failure (HF) was first described as an emerging epidemic in approximately the mid-1990s. Since then, the number of HF patients increased globally, mainly due to the ageing population and improved treatment in the field of cardiovascular diseases [1,2]. HF still remains a global public health issue affecting more than 37.7 million people worldwide [3,4]. HF incidence in Europe and the USA ranges widely from 1 to 9 cases per 1000 person-years, and depends on the sample studied and the diagnostic criteria used [1]. HF involves several restrictions in a patient's daily life, mostly attributed to the cognitive and physical impairment that accompanies this syndrome. Accordingly, HF implies personal, family, social and economic consequences that adversely affect a patient's quality of life [5,6].

Self-care is a fundamental element in HF treatment which keeps in pace with patients in the trajectory of this debilitating chronic disease. Self-care is defined as the activities undertaken by individuals with the aim of enhancing or restoring health and preventing or limiting illness. Self-care, as a dynamic and continuous process, includes three inseparable dimensions. The first is self-care maintenance, which captures behaviors referred as treatment adherence. The second is symptom perception, which involves the recognition and interpretation of symptoms. The third is self-care management, which includes decision making and response to symptom aggravation [7].

HF patients need to make decisions on a daily basis in order to respond effectively to potential exacerbation and prevent hospitalizations, readmissions or outpatient visits [6–8]. Interestingly, adverse clinical events in HF are less likely to occur in patients who seek consultation at the onset of symptoms and practice appropriate self-care behaviors [4]. Unfortunately, a large proportion of HF patients show low self-care behavior levels or an inadequate response to the required self-management interventions [6–8]. Self-care through decision-making is based on how patients' feel rather than on clinical indicators of worsening symptoms [8]. Patients' views, preferences and needs are demonstrated as important dimensions of a tailored self-care advice in HF [9–11].

Health care professionals are able to plan effective interventions only after assessing patients' perceptions about self-care. Maximizing self-care means minimizing healthcare expenditure and improving their quality of life. Accordingly, the notable aspect is to comprehend the importance of self-care as perceived by the patients. The aim of this study was to conduct a validity and reliability analysis of the new 14-item IPSC scale, which measures how important HF patients consider their physical self-care behavior (IPSC, Importance of Physical Self-care).

## 2. Materials and Methods

### 2.1. Design, Setting, and Period of the Study

In this study, 52 hospitalized HF patients who suffered from chronic heart failure were enrolled during the period January–March 2021. This was a cross-sectional descriptive study. Participants were selected using the method of convenience sampling.

### 2.2. Inclusion and Exclusion Criteria of the Sample

The inclusion criteria of the sample were (a) age above 18 years; (b) the ability to write, read and understand the Greek language; and (c) the ability to read and sign the informed consent form. The exclusion criteria included patients with (a) a history of mental illness; (b) with cognitive impairment and sight or hearing problems, and patients hospitalized due to other comorbidities and not HF; and (c) those with acute heart failure.

### 2.3. Data Collection and Procedure

Collection of data was performed using the method of interview using a specially designed research instrument. It was carried out in afternoons, when participants had no tasks or laboratory tests to perform. The procedure lasted approximately twenty minutes. Subjects were first interviewed the second day of hospitalization in order to evaluate their needs when they were at home and not under the supervision of health care professionals.

### 2.4. Research Instrument

The instrument used was a questionnaire, which included patients' characteristics and the 14-item IPSC scale, which measures how important HF patients consider their physical self-care behavior.

### 2.5. Measuring Importance of Physical Self-Care Behavior

The 14-item IPSC scale was created by the researchers in order to assess how important the HF patients considered their physical self-care behavior (Appendix A Table A1). These items were selected taking into account the questionnaire "Needs of hospitalized patients

with coronary artery disease" [11] and previous literature reviews [6,8,12]. More in detail, patients declared how important they considered each item of physical self-care behavior included in the scale to be. The following answers for each item corresponded to the 4-point Likert scale: not at all (score 4), a little (score 3), a lot (score 2) and very much (score 1). The total score (sum) took a possible range of values of 14–56. Higher scores indicated a low importance of self-care. The 14-item IPSC scale did not include reverse-scored items.

Since it was a new scale created by the researchers, the necessary reliability and validity analysis of the tool was carried out. Immediately after its design, the IPSC scale was completed by five patients in order to determine whether the items were considered clear, understandable and in a logical order (face validity). In addition, the same patients along with five health professionals who had long-term experience in the field of HF were asked to judge the content of the scale (content validity). All considered the items highly representative of measuring the importance of physical self-care behavior.

### 2.6. Ethical Considerations

The present study was approved by the Research Committee of the two public hospitals. Patients who met the entry criteria were informed by the researcher for the purposes of this study. All patients participated in the study after they had given their written consent. Data collection guaranteed anonymity and confidentiality. All subjects had been informed of their rights to refuse or discontinue participation in the study, according to the ethical standards of the Declaration of Helsinki (1989) of the World Medical Association.

### 2.7. Statistical Analysis

The construct validity was assessed by comparing the score of each item with the total score of the scale on the importance of self-care. The comparison was performed using the Spearman's rho correlation coefficient. The index takes values between −1 and +1. Values close to +1 indicate high construct validity.

The internal consistency of the scale (reliability) was assessed by calculating the Cronbach Alpha index. The value of this index ranges from 0 to –1. Large values of the Alpha index indicate a high coherence of the questions that constitute the scale (reliability). The Cronbach Alpha index was used to identify questions that reduced the internal consistency of the questionnaire and, therefore, had to be excluded.

The repeatability test (test–retest) was performed by applying the statistical criterion intraclass correlation coefficient (ICC). To perform the repeatability test, the participants completed the scale a second time 14 days after the first completion. This criterion's values range between −1 and +1. Values close to +1 indicate high repeatability of the scale. Results for the repeatability test are presented with the ICC index and 95% confidence intervals (CI).

Discriminant validity of the scale was assessed using basic statistical criteria to compare final scores between patient characteristics. The statistical criteria used were the independent samples t-test and the ANOVA criterion.

Finally, total scores are presented with the mean and standard deviation, as well as the median and interquartile range. Patient characteristics are presented with absolute and relative frequencies (%). The observed significance level of 5% was considered statistically significant. All statistical analyzes were performed with SPSS version 25 (SPSS Inc., Chicago, IL, USA).

## 3. Results

### 3.1. Sample Description

The majority of participating patients were women (55.8%), over 70 years old (34.6%), were of NYHA class III (48.1%), lived in Attica (69.2%), suffered this illness from 6–10 years (32.7%) and were sufficiently informed about HF (44.2%) (Table 1).

**Table 1.** Sample Description (N = 52).

|  | **N (%)** |
|---|---|
| **Gender** |  |
| Male | 23 (44.2%) |
| Female | 29 (55.8%) |
| **Age (years)** |  |
| 30–40 | 1 (1.9%) |
| 41–50 | 4 (7.7%) |
| 51–60 | 12 (23.1%) |
| 61–70 | 17 (32.7%) |
| 71–80 | 18 (34.6%) |
| **NYHA** |  |
| I | 1 (1.9%) |
| II | 13 (25.0%) |
| III | 25 (48.1%) |
| IV | 13 (25.0%) |
| **Residency** |  |
| Attica | 36 (69.2%) |
| Capital City | 6 (11.5%) |
| Small Town | 4 (7.7%) |
| Village | 6 (11.5%) |
| **Years suffering the disease** |  |
| <1 | 6 (11.5%) |
| 2–5 | 11 (21.2%) |
| 6–10 | 17 (32.7%) |
| 11–15 | 10 (19.2%) |
| >15 | 8 (15.4%) |
| **Informed about HF** |  |
| Very | 9 (17.3%) |
| Enough | 23 (44.2%) |
| A little | 15 (28.8%) |
| Not at all | 5 (9.6%) |

*3.2. Scores of the 14 Item IPSC Scale*

Descriptive statistics of the IPSC scores are presented in Table 2. Comparing means and medians with the possible range of scores, regarding the importance of physical self-care behavior, the patients assessed it as moderately important (mean 27.9 ± 4.9 and median 29).

**Table 2.** 14 item IPSC scale (N = 52).

|  | **No. of Questions** | **Score Range** | **Mean (SD)** | **Median (IQR)** |
|---|---|---|---|---|
| **Importance of physical self-care behavior** | 14 | 14–56 | 27.9 (4.9) | 29 (24.5–31) |

SD: Standard Deviation, IQR: Interquartile Range.

*3.3. Construct Validity*

Table 3 presents the results for the construct validity. All sub-items of the IPSC scale were statistically significantly correlated to the total score (*p*-values < 0.05) with correlation coefficients of rho > 0.250, indicating moderate-to-strong correlations. This meant that all items were important in the calculation of the final score.

**Table 3.** Construct validity for 14 item IPSC scale.

| How Important Are the Following Items for You? | Total Score of Importance of Physical Self-Care Behavior | |
|---|---|---|
| | Rho | *p*-Value |
| 1. Be aware of food restrictions myself | 0.348 | **0.011** |
| 2. Be aware of fluid intake restrictions myself | 0.680 | **<0.001** |
| 3. To make sure that I remain active | 0.626 | **<0.001** |
| 4. To make sure that I take my medicines every day | 0.285 | **0.041** |
| 5. Controlling my weight (increase or decrease) | 0.666 | **<0.001** |
| 6. Measuring my own blood pressure, pulse and respirations | 0.432 | **0.001** |
| 7. Make sure I get enough rest and sleep | 0.348 | **0.011** |
| 8. Keeping a diary of the progress of my symptoms | 0.680 | **<0.001** |
| 9. To take the necessary actions, upon suspicion of symptom aggravation | 0.356 | **0.010** |
| 10. To take care of my condition based on my personal needs e.g., fatigue | 0.585 | **<0.001** |
| 11. To make sure that I adjust my daily life according to my physical state | 0.586 | **<0.001** |
| 12. To strengthen my abilities or behaviors to limit my symptoms | 0.607 | **<0.001** |
| 13. To feel that I am contributing to the improvement of my symptoms | 0.364 | **0.008** |
| 14. To arrange follow up in hospital that monitors me and takes into account my HF history | 0.484 | **0.001** |

### 3.4. Reliability: Internal Consistency

Table 4 presents the results of the internal consistency (reliability). The internal consistency of the items that constitute the total score was found high (Cronbach's a > 0.7), which indicates high reliability of the participants' declarations.

**Table 4.** Reliability:Internal Consistency.

| | Cronbach's a |
|---|---|
| **Importance of physical self-care behavior** | 0.795 |

### 3.5. Test–Retest

Table 5 presents the results of the repeatability test. It was found that all scores had high repeatability, which indicates a high reliability of the participants' declarations (*p*-values < 0.001 and ICCs > 0.7).

**Table 5.** Test–retest.

| | ICC (95% CI) | *p*-Value |
|---|---|---|
| **Importance of physical self-care behavior** | 0.805 (0.683–0.883) | 0.01 |

### 3.6. Discriminant Validity

Table 6 presents the scores of the items with respect to patients' demographic characteristics (gender, age, place of residence), clinical characteristics (year of disease, NYHA class), and their degree of knowledge about their disease. A statistically significant association was observed between the importance of physical self-care behavior and years suffering the illness (*p* = 0.007), as well as NYHA class (*p* = 0.030). More specifically, physical self-care behavior was considered more important by patients who suffered the illness for less than 5 years (mean value 24.6) and those of NYHA class I–II (mean value 24.9). The remaining characteristics were not found to be significantly associated with the scores.

**Table 6.** Discriminant validity.

| | Importance of Physical Self-Care Behavior |
|---|---|
| | **Mean (SD)** |
| **Gender** | *p* = 0.057 |
| Male | 26.3 (5.7) |
| Female | 29.3 (3.9) |
| **Age (years)** | *p* = 0.106 |
| ≤60 | 28.2 (3.3) |
| 61–70 | 25.6 (5.1) |
| >70 | 29.9 (5.3) |
| **Residency** | *p* = 0.796 |
| Attica | 27.9 (5.4) |
| Other | 27.9 (3.9) |
| **Years suffering the disease** | ***p* = 0.007** |
| <5 | 24.6 (5.0) |
| 6–10 | 29.6 (4.5) |
| >10 | 29.5 (3.7) |
| **NYHA** | ***p* = 0.030** |
| I–II | 24.9 (5.4) |
| III | 28.4 (4.2) |
| IV | 30.4 (4.2) |
| **Informed about disease** | *p* = 0.578 |
| Very/Enough | 28.1 (4.9) |
| A little/Not at all | 27.7 (5.1) |

SD: Standard Deviation.

## 4. Discussion

Scales are the most widely used tools in clinical practice, largely due to their low cost and ease of application. To the best of our knowledge, the measurement of how important patients perceive self-care is limited. Additionally, regarding the importance of self-care, there is no gold standard which has been thoroughly tested and has a reputation in the HF field as a reliable method. The present 14-item IPSC scale is a tool applied within a short time. It is a reliable instrument that presents satisfactory validity, making it suitable for use in research and clinical settings. The 14-IPSC scale can be used both in interview settings and as a self-reporting instrument. Furthermore, it can be completed at the same time with the medical record at hospital admission or at discharge. When applying this tool, health care professionals obtain a prompt assessment regarding the level of importance that patients pay to their physical self-care behavior.

HF treatment involves medication, adoption of nutritional recommendations (mainly sodium and fluid restrictions), regular physical exercise, lifestyle modifications, symptoms monitoring (increase of body weight, dyspnea), refilling of prescriptions and frequent reassessment. In HF, the ultimate goal of self-care is to maintain stability and prevent avoidable complications or rehospitalizations [13–16].

However, the crucial point is "do patients comprehend the importance of self-care?". A noticeable finding of the present study is that HF patients evaluated physical self-care behavior as moderately important. In light of this finding, it is imperative to early screen this vulnerable group of patients and afterwards refer them to educational interventions that emphasize their ability to perform activities. One aspect is the development of a patient-centered system that enhances self-care through telephone consultations and support for these persons. It is widely accepted that low self-care is not an uncommon phenomenon in HF patients. A key challenge confronting investigators is to discern whether patients underestimate the importance of self-care due to several complex reasons, such as cognitive impairment, psychiatric disorder and social isolation.

Various factors are held to be responsible for considering self-care as an issue of low importance. For instance, patients often report to "know" self-care recommendations but at the same time experience an inability in "how" to apply this knowledge in their

day-to-day lives [8,9]. Furthermore, patients may rank the need of receiving medication of low importance because they show little confidence in the beneficial effects of medication on functional capacity or symptoms [17]. Furthermore, HF patients perceive exercise as more difficult than other recommendations such as diet, medication, smoking cessation and reassessment appointments. Psychological factors increase difficulties in adherence to exercise by reducing interest and motivation. Patients may show a reluctance to start physical exercise due to lack of skills or fear of physical activity with a "bad heart" [18,19]. Moreover, low health literacy may partially explain why patients do not recognize the important role of self-care. Interestingly, low health literacy is associated with diminished self-care knowledge and behavior, reduced use of preventive health services, frequent readmissions and low self-efficacy [20–22].

When attitudes about a specific self-care behavior are favorable, for example, following a low-salt diet (thought to be easy to do or associated with a good outcome), one is more likely to engage in self-care. Patients' values influence their choices about health, including HF self-care. The effects of HF on daily life, including symptoms, may influence behavior prioritization. When interventions addressing knowledge and skills are unsuccessful, patients' values and perceptions need to be addressed. Social norms and cultural beliefs influence a patient's willingness to adopt self-care behaviors. Experience, knowledge, skill and values contribute to HF self-care decisions but are unique, so people prioritize them differently according to the context in which these decisions are made [13,23].

In the present study, self-care was considered more important by patients who suffered the illness for less than 5 years and those of NYHA class I–II. This finding seems to be inconsistent with the theoretical model of HF self-care, which underlines the naturalistic decision-making. According to this model, the effectiveness of self-care depends on the knowledge which is developed or accumulated through experience [8,24]. Longer HF duration is a determinant of higher self-care [4,24]. On the other hand, years of suffering this clinical syndrome reflects the symptoms' severity, such as dyspnea at rest or on exertion and fatigue which is the most prevalent symptom in HF patients, ranging from 69% to 88% [25,26]. Additionally, in advanced NYHA classes, HF patients experience a loss of functional independence in daily activities such as feeding, dressing, housekeeping, bathing and walking [25]. Moreover, functional capacity worsens as age advances [27]. Given that HF is predominately a disease of the elderly, it is assumed that patients who experience physical decline have the tendency to consider self-care of low importance.

Gaps or misconceptions about the importance of self-care ought to be addressed in patient–provider discussions. Clinical approaches will be most effective when tailored to patients' profiles. Providing information is a key element that promotes decision making and encourages patient participation in the treatment process [5,28,29].

*Limitations of the Study*

The limitations of this study include the cross-sectional design and the use of self-reporting instruments. Furthermore, convenience sampling is one of the limitations, as this method is not representative of all populations with HF living in Greece, thus restricting the generalizability of the results. Moreover, it is important to consider other confounders that were not a subject of inquiry in the present study but have been shown to have an effect on self-care such as cognitive impairment, depression and isolation. Additionally, there was no instrument used as a gold standard which had been thoroughly tested and has a reputation in HF field as a reliable method.

**5. Conclusions**

The IPSC scale may help health professionals to gain a deeper understanding of patients' self-care behavior. Moreover, measuring the importance that HF patients attribute to their physical self-care behavior may alert clinicians to areas that would otherwise be overlooked.

Furthermore, exploring the importance that HF patients invest in self-care is essential for health services. For example, it provides an insight into the essential meanings of this population or is critical when designing education or other interventions with the ultimate goal to improve the position of HF patients within the context of self-care.

Hence, it is relevant to know which patient-related characteristics (years suffering this illness, NYHA) are prognostically important, i.e., have true predictive value for failure to self-care, especially when they consider it of low importance. Enhancing self-care will markedly decrease the economic, medical, individual and social burden of HF. The present findings have the potential to stimulate and guide future research efforts.

**Author Contributions:** A.T., supervision, project administration, writing—review and editing, and conceptualization; I.K., investigation; methodology and writing—original draft; G.G., writing—review and editing and methodology; N.P., data curation, visualization and software; A.K., validation and formal analysis; T.K., investigation and resources; G.T., visualization and writing—review and editing; M.P., writing—original draft, data curation, methodology, supervision and writing—review and editing. All authors have read and agreed to the published version of the manuscript.

**Funding:** This research study received funding from the Special Account for Research Grants of the University of West Attica, Greece.

**Institutional Review Board Statement:** The present study was conducted according to the ethical standards of the Declaration of Helsinki, and approved by the Research Committee of the hospitals.

**Informed Consent Statement:** Informed consent was obtained from all participants involved in the study.

**Conflicts of Interest:** All authors declare no conflict of interest in this paper.

## Appendix A

**Table A1.** 14 item IPSC scale.

| 14 Item IPSC Scale | | | | |
|---|---|---|---|---|
| **How Important Are the Following Items for You?** | **Not at All** | **A Little** | **A Lot** | **Very Much** |
| 1. Be aware of food restrictions myself | | | | |
| 2. Be aware of fluid intake restrictions myself | | | | |
| 3. To make sure that I remain active | | | | |
| 4. To make sure that I take my medicines every day | | | | |
| 5. Controlling my weight (increase or decrease) | | | | |
| 6. Measuring my blood pressure, pulse and respirations | | | | |
| 7. Make sure I get enough rest and sleep | | | | |
| 8. Keeping a diary of the progress of my symptoms | | | | |
| 9. To take the necessary actions, upon suspicion of symptom aggravation | | | | |
| 10. To take care of my condition based on my personal needs, e.g., fatigue | | | | |
| 11. To make sure that I adjust my daily life according to my physical state | | | | |
| 12. To strengthen my abilities or behaviors to limit my symptoms | | | | |
| 13. To feel that I am contributing to the improvement of my symptoms | | | | |
| 14. To arrange follow up in hospital that monitors me, and takes into account my HF history | | | | |

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
