# Peer review of "Measuring Importance of Physical Self-Care Behavior in Patients with Heart Failure: Validation and Reliability Analysis of 14-item IPSC Scale"

_clinpract, doi:10.3390/clinpract13020032_

Round 1
Reviewer 1 Report
This is a very interesting study that attempts to quantify the extent to which patients themselves value the importance of physical self-care behaviors in heart failure patients.
However, I consider this paper unsuitable for publication in the following respects.
First, it does not specify when the subjects were first interviewed. It would be expected that the importance of physical self-care behaviors would change depending on the length of hospital stay for hospitalized heart failure patients. In addition, although the subjects were interviewed again after 14 days, the 14-day course of hospitalized heart failure patients may have a significant impact on how important they consider physical self-care behaviors to be. Therefore, I believe that they are not appropriate subjects for this study. I believe that the inclusion of patients with chronic heart failure would be more in line with the purpose of this study.
I also believe that there is a lack of validation regarding construct validity. You may wish to refer to the following article regarding validity.
Messick, Samuel (1989), “Validity,” in Educational Measurement, ed. Robert L. Linn, New York: Macmillan Publishing Company, 13-104.
Messick, Samuel (1995), “Validity of Psychological Assessment: Validation of Inferences from Persons' Responses and Performances as Scientific Inquiry into Score Meaning,” American Psychologist, 50(9), 741-49.
Author Response
Reviewer no 1
- this is a very interesting study that attempts to quantify the extent to which patients themselves value the importance of physical self-care behaviors in heart failure patients.
Answer : This article explores the way patients’ perceive the importance of their needs. Our ultimate goal was to understand the way patients understand their illness in order to create and individualized care and approach
- However, I consider this paper unsuitable for publication in the following respects. First, it does not specify when the subjects were first interviewed. (I mentioned in method and material)
Answer : The patients were firstly interviewed the second day of hospitalization in order to evaluate their needs when they were at home
- It would be expected that the importance of physical self-care behaviors would change depending on the length of hospital stay for hospitalized heart failure patients. In addition, although the subjects were interviewed again after 14 days, the 14-day course of hospitalized heart failure patients may have a significant impact on how important they consider physical self-care behaviors to be.
Answer : The significance of needs refers prior to hospitalization ! we measure 2nd time for test -retest. Noteworthy, in a busy general hospital, health care professionals tend to focus on the biological dimension of the disease. Possibly, harder “work” is needed while in hospital (i.e education interventions in order to discover patients’ perceptions and enhance self care. Secondly, Other dimension might be responsible for the “no change depending on the length of hospital stay” such as stress during hospital, cognitive impairment BUT these were not the subject of the present research (:possibly in a future research). Another possible explanation is that heart failure patients may rely on health professionals while at hospital and these patients need to be screened and be a target of future interventions.
- Therefore, I believe that they are not appropriate subjects for this study. I believe that the inclusion of patients with chronic heart failure would be more in line with the purpose of this study. It REFERS to chronic heart failure and I just wrote it
Answer : This is corrected within the text
- I also believe that there is a lack of validation regarding construct validity. You may wish to refer to the following article regarding validity.
Messick, Samuel (1989), “Validity,” in Educational Measurement, ed. Robert L. Linn, New York: Macmillan Publishing Company, 13-104.Messick, Samuel (1995), “Validity of Psychological Assessment: Validation of Inferences from Persons' Responses and Performances as Scientific Inquiry into Score Meaning,” American Psychologist, 50(9), 741-49.
Answer : Sorry but we can not understand what the reviewer needs. In this case, we apologize but we cannot any take any further step
Reviewer 2 Report
The manuscript entitled “Measuring importance of physical self-care behavior in patients with heart failure: Validation and Reliability Analysis of 3 IPSC-14 Item Scale” is an interesting study, which not only stresses the importance of self-care for heart failure patients, but also introduces an important tool (IPSC-14 Item Scale) based on the recognizing the importance of self-care by patients. By validating the efficiency of this scale in multiple clinical centers, it was tested that this scale was with high efficiency and consistency, the data obtained was reliable and associated with the degrees of heart failure and the years of suffering this method. By completed 14 questions by the patients, it is prompt for clinicians and patients to quantify the importance of self-care in the personalized treatment.
Here are some questions and recommendations,
1. The questions in the scale, all of the questions are quite subjective, is there any possibility that some objective questions, regarding the symptoms, such as the degrees of edema, exercise tolerance, etc. can be added to this scale? It might be more persuasive and accurate in evaluating the need or requirement of self-care from the patients.
2. In the statistical part, the authors mentioned that “the statistical criteria used were 135 the independent sample t-test and the ANOVA criterion.” Please indicate which analysis were used in which settings.
3. Another limitation of this scale is that it only adapts to patients who have normal recognition ability, normal mental status, and well-educated background to some extent, that they can comprehend the scale and makes the proper evaluation; However, for patients who have mental disorder or recognition problems, poor education background, it would not be feasible. Will there be some alternatives for these patients? Thanks.
Author Response
Here are some questions and recommendations,
- The questions in the scale, all of the questions are quite subjective, is there any possibility that some objective questions, regarding the symptoms, such as the degrees of edema, exercise tolerance, etc. can be added to this scale?
Answer : Of COURSE, if you have noticed we: A. included the New York Heart Association Functional Classification(NYHA scale) where doctors usually classify patients' heart failure according to the severity of their symptoms that limit their daily life and B. explored the severity of the disease according to the years suffering heart failure. Both variables (NYHA and years) were statistically significantly associated with the scale.
We did not explore the level of dyspnea, orthopnea and exercise tolerance (through measuring VO2max ) because placed our focus on patients’ perceptions. Only when, we deeply understand what patients perceive as important then we ensure patients’ participation in the therapeutic regimen!
- It might be more persuasive and accurate in evaluating the need or requirement of self-care from the patients.
Answer : Of course self care is the cornerstone in heart failure treatment and includes self maintenance and self management. I firmly believe that first we evaluate patients perceptions, then we target to their “weak” points and finally we use this knowledge to enhance self care in order to minimize health care cost and improve their quality of life.
- In the statistical part, the authors mentioned that “the statistical criteria used were 135 the independent sample t-test and the ANOVA criterion.” Please indicate which analysis were used in which settings.
Answer : This is corrected within the text
- Another limitation of this scale is that it only adapts to patients who have normal recognition ability, normal mental status, and well-educated background to some extent, that they can comprehend the scale and makes the proper evaluation; However, for patients who have mental disorder or recognition problems, poor education background, it would not be feasible. Will there be some alternatives for these patients? Thanks.
Answer : These patients are excluded as it is referred in exclusion criteria
Round 2
Reviewer 1 Report
何も言うことはない。